# Surface Modification of Graphene for Use as a Structural Fortifier in Water-Borne Epoxy Coatings

Ya Liu [1,2], Chao Xia [3], Abdelhafid Zehri [1], Lilei Ye [4], Nan Wang [4], Boris Zhmud [3], Hongbin Lu [2,*] and Johan Liu [1,5,*]

1   Electronics Materials and Systems Laboratory (EMSL), Department of Microtechnology and Nanoscience (MC2), Chalmers University of Technology, Kemivägen 9, SE-412 96 Göteborg, Sweden; yaliu@chalmers.se (Y.L.); hafid@chalmers.se (A.Z.)
2   State Key Laboratory of Molecular Engineering of Polymers, Department of Macromolecular Science, Collaborative Innovation Center of Polymers and Polymer Composites, Fudan University, 2005 Songhu Road, Shanghai 200433, China
3   Applied Nano Surfaces AB, Knivstagatan 12, SE 753 23 Uppsala, Sweden; chao.xia@appliednanosurfaces.com (C.X.); boris.zhmud@appliednanosurfaces.com (B.Z.)
4   SHT Smart High Technology AB, Kemivägen 6, SE 412 58 Gothenburg, Sweden; lilei.ye@sht-tek.com (L.Y.); Nan.wang@sht-tek.com (N.W.)
5   SMIT Center, School of Mechanical Engineering and Automation, Shanghai University, No. 20, Chengzhong Road, Shanghai 201800, China
*   Correspondence: hongbinlu@fudan.edu.cn (H.L.); johan.liu@chalmers.se (J.L.)

**Abstract:** Graphene, the typical two-dimensional sp$^2$ hybridized carbon allotrope, is widely used as a filler for improving the mechanical performance of polymers. However, its superhydrophobic surface makes it a big challenge to obtain stable graphene dispersions, especially in water-borne systems. On the contrary, graphene oxide (GO) shows excellent dispersibility in water, but strong oxidants and acids destroy its structure and degrade its mechanical properties. This largely limits its application in water-borne coatings. In this work, graphene from mechanical exfoliation was surface modified by p-aminophenol derived diazonium salt to achieve a homogenous dispersion. Moreover, the hydroxyl groups in p-aminophenol are able to combine with epoxy resins during the curing process to improve mechanical performance of the final coatings. The result shows that functionalized graphene shows a lower coefficient of friction and better abrasion resistance compared to GO.

**Keywords:** functionalized graphene; water-borne epoxy; tribological properties

## 1. Introduction

Epoxy resins (ER) are an important engineering material featuring outstanding corrosion resistance, high mechanical strength, and strong adhesion to heterogeneous substrates [1,2]. ERs can be used as adhesives, protection coatings, composites, laminating materials, etc. In recent years, a tremendous interest has been raised in scientific and industrial communities to apply epoxy composites as sliding components, aiming to increase the product service life and performance.

Sliding, rolling, or rotating contacts ubiquitous in a variety of manmade systems all generate friction. If the friction is not efficiently controlled, it often leads to high wear, shorter service life, and reduced reliability [3]. In this regard, the deposition of low-friction coatings, such as high-strength epoxy coatings, has become an effective strategy for extending the service life of coated parts. Considering the environmental issues and the improvement in tribological properties, water-borne epoxy coatings are a promising option [4]. Nowadays, for improving wear-resistance of water-borne coatings, an effective

and widely accepted method is to use nano-fillers, such as $SiO_2$ [5], nanodiamonds [6], carbon nanotubes [7], graphene [8], graphene oxide, [9] etc.

As a typical 2D material, graphene offers unique friction and wear properties that are not usually observed in conventional materials [10]. Besides its exceptional thermal, electrical, optical, and mechanical properties, graphene can also serve as a solid lubricant. The high chemical inertness and excellent shear-slide capability make graphene a promising alternative for constructing wear-resisting coating layers [11]. Ultrathin graphene layers, even in multilayer configurations, can be applied to nano- or microsystems such as microelectromechanical systems (MEMS) and nanoelectromechanical systems (NEMS) with oscillating, rotating, and sliding contacts to reduce friction and wear [12].

When used in water-borne coating systems, the primary challenges is to produce stable homogenous graphene dispersions. To address this issue, surface modifications, involving covalent and noncovalent hydroxyl and carboxyl functionalization, have been performed on the surface of graphene through the edges or defect sites to improve graphene's dispersibility [3,13]. Besides that, epoxy groups are able to react with the functional groups grafted onto graphene, including hydroxyl, amino, etc., [14] to build a stable crosslinked network between functional graphene and an epoxy binder [15]. Such crosslinking is favorable for the mechanical performance of graphene-filled composites.

In this work, a diazonium salt with hydroxyphenyl groups was used to modify the surface of graphene. The hydroxyl groups and benzene rings in the hydroxyphenyl moiety help adjust the surface hydrophilicity of graphene: The functionalized graphene can be readily dispersed in water to form a stable dispersion. On the other hand, the hydroxyl groups are able to further react with the epoxy groups to build a strong bonding between the graphene flakes and the epoxy binder, significantly improving the tribological properties of the epoxy coating. Compared with graphene oxide (GO), the functionalized graphene (FG) sample exhibits a lower coefficient of friction and higher abrasion resistance.

## 2. Materials and Methods

### 2.1. Preparation of Graphene (G)

Graphene powder was purchased from Carbonene (Product XTG-P-0762, Deyang, China). GO was purchased from Applied Graphene Materials (Product A-GNP10, Redcar, England). The p-aminophenol and sodium hydroxide were purchased from Sigma-Aldrich (Saint Louis, MO, USA) with purity <98%. The 37% hydrochloric acid and sodium nitrite (for analysis) were purchased from Sigma-Aldrich (Saint Louis, MO, USA). All chemicals were used without additional purification.

### 2.2. Preparation of Diazonium Salt (DS)

By analogy with the Sandmeyer reaction, 2 g *p*-aminophenol was added to 100 mL 5% sodium hydroxide solution at 50 °C and stirred for 30 min. Then 20 mL 0.1 M sodium nitrite solution was added to the dispersion. Subsequently, the dispersion was transferred to an ice-water bath with a temperature of about 0–4 °C. In the next step, 15 mL 1 M hydrochloric acid was added into the dispersion and stirred for 30 min. Afterwards, centrifugation was used to isolate the product. The yield of *p*-hydroxybenzene diazonium chloride was 80%. Note that due to its tendency to self-decomposition, the diazonium salt should be prepared immediately before the surface modification.

### 2.3. Preparation of Functionalized Graphene

First, 0.5 g graphene powder was dispersed into 100 mL distilled water in an ultrasonic bath. Then pH value was adjusted to 10 by adding the sodium hydroxide solution. After that, 2 g 5 wt. % diazonium salt solution was gradually added to the graphene dispersion under mechanical stirring for 8 h and continuous cooling using the ice-water bath. In the end, centrifugation was used to remove excess chemicals and impurities from the solution. The yield of modified graphene was 72% based on graphene weight.

### 2.3.1. Preparation of Water-Borne Coatings Containing Functionalized Graphene

Water-borne epoxy coatings were formulated by Applied Nano Surfaces Sweden AB (Uppsala, Sweden). Typically, 1 g functional graphene (FG) powder was dispersed in 30 mL distilled water under ultrasonification. Subsequently, the FG dispersion was added to the commercial Tricolit TSC coating base with mechanical stirring for 30 min. Next, the obtained coating slurry was sprayed onto steel plates with a spray gun to achieve a dry film with a thickness between 15 to 25 μm. The sprayed plates were dried at room temperature for 30 min, and then transferred to a furnace with a temperature profile of 50 °C for 30 min, 90 °C for 1 h, 120 °C for 1 h, and 250 °C for 0.5 h for final curing.

### 2.3.2. Characterization

Fourier transform infrared spectroscopy (FT-IR, PerkinElmer, UATR Two, 4000–500 cm$^{-1}$, Waltham, MA, USA) and X-ray photoelectron spectroscopy (XPS, PHI 5000 VersaProbe III, monochromatic Al $K\alpha$ X-ray, Kanagawa, Japan) were employed to identify the chemical structures of graphene (G) and functionalized graphene (FG). Thermalgravimetric analysis (TGA, PerkinElmer, Pyris 1, Waltham, MA, USA) was applied to determine the mass ratio of functional groups in FG from 60 to 600 °C under nitrogen atmosphere. Transmission electron microscopy (TEM, FEI Tecnai, T20, 300 KV, Hillsboro, OR, USA) was used to observe the morphologies of G and FG. The morphology of the epoxy coating film was analyzed by scanning electron microscopy (SEM, Zeiss Supra 60 VP, Oberkochen, Germany) with acceleration voltage 6–12 KV. Raman spectra were obtained with LabRAM HR-800 (Horiba, Kyoto, Japan) with 514 nm laser and ×100 objective.

### 2.3.3. Tribological Tests

Friction tests were performed using a reciprocating ball-on-disc test rig (Figure 1). The 100Cr6 steel balls with Ø11 mm and Ra 0.065 μm were used during the tests. Dry abrasion tests were conducted for 30 min using a constant velocity of 200 rpm and stroke length of 10 mm under dry condition, with a load of 4 N. Lubricated friction tests were conducted using velocity ramps of 0 to 100 to 0 rpm (10 circles), stroke length of 10 mm, and a load of 3 N. Castrol Edge 0W-40 motor oil (Castrol, Germany) was used as the lubricant. Wear tests were carried out in a customized rig (see Figure 2c) resembling the industry-standard Taber abraser (ASTM D-4060). The tests were used for the relative ranking of wear resistance (one coating vs. another). A nylon ring was rotating with a speed of 130 rpm and a load of 1 N against the coating on the stainless-steel substrate. The time required for total wear out of the coating was recorded: The longer coating breakdown time, the better the abrasion resistance was.

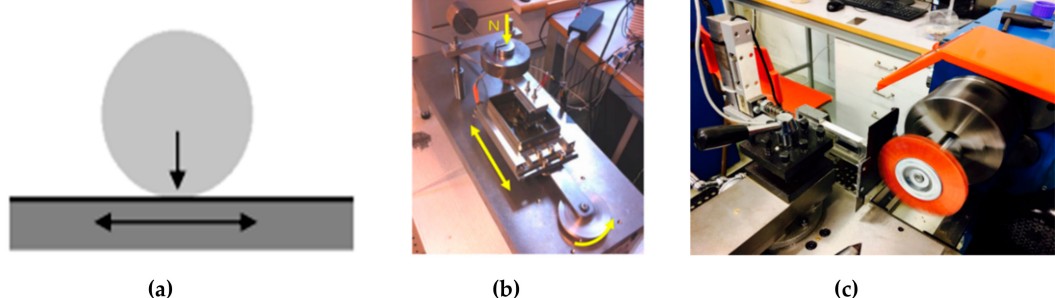

| **(a)** | **(b)** | **(c)** |

**Figure 1.** (**a**) Ball-on-flat friction and wear test setup, (**b**) tribological test rig, (**c**) abrasion test rig.

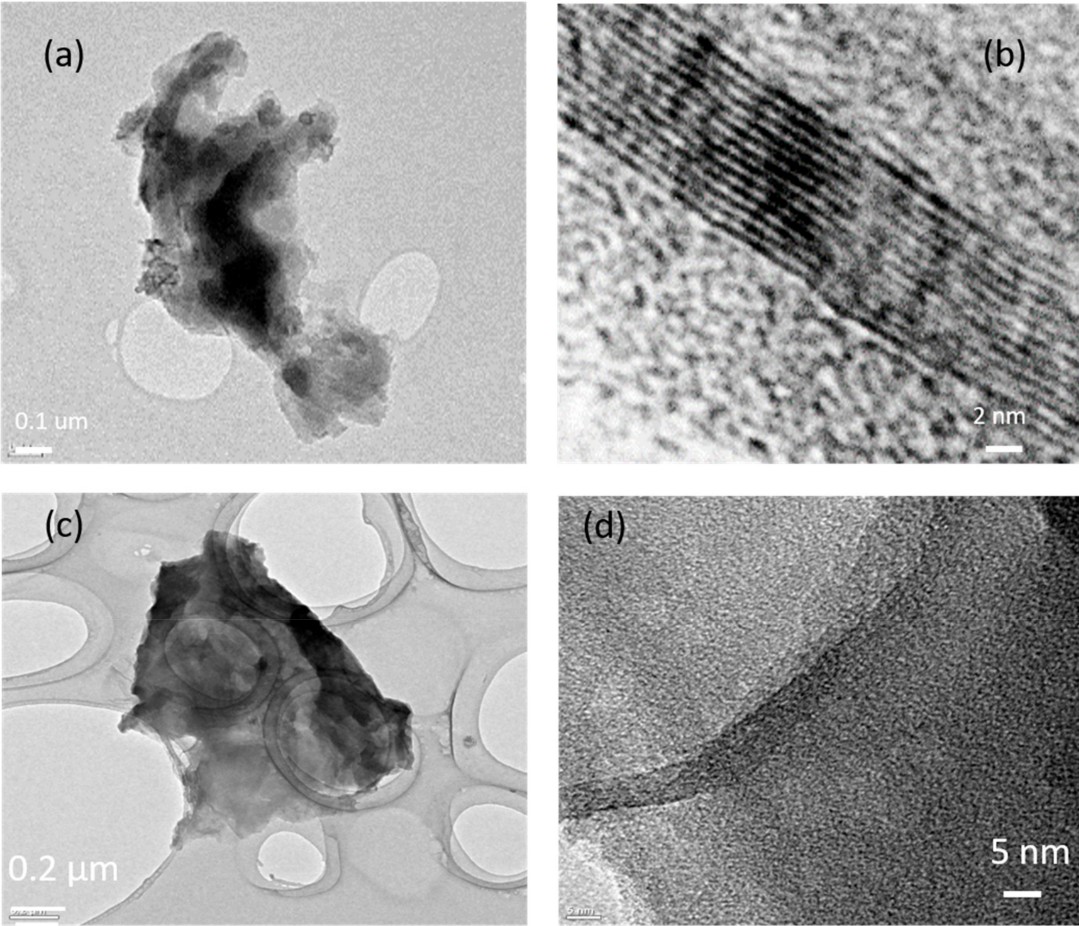

**Figure 2.** TEM images of graphene (G) and functionalized graphene (FG). (**a**) Low-resolution and (**b**) high-resolution TEM images of G; (**c**) low-resolution and (**d**) high-resolution TEM images of FG.

## 3. Results and Discussion

### 3.1. Characterization of Functional Graphene

TEM was used to observe the influence of functionalization on graphene morphology. As shown in Figure 2, the pristine graphene sheets consist of approximately 10 graphene layers (see Figure 2b). After functionalization, a new grafted layer was observed on the surface of graphene (see Figure 2d). In the high-resolution TEM image of FG, its crystal structure was not as sharp as graphene. This may be due to the functional groups attached to the graphene surface.

FTIR spectroscopy was applied to determine the functional groups in pristine graphene and FG. As shown in Figure 3a, a small peak at 1201 cm$^{-1}$ corresponding to C–O stretching vibrations appears at the FTIR spectrum of pristine graphene, indicating the presence of a small amount of oxidized sites. A peak at 2120 cm$^{-1}$ corresponds to the N≡N stretching vibrations in aryl diazonium salts. After functionalization with the prepared DS, the peak at 2120 cm$^{-1}$ disappears. Other indicative peaks are at 1191 cm$^{-1}$ due to C–O stretching, and at 1730 cm$^{-1}$ due to C=O stretching. Further, the peak at 1061 cm$^{-1}$ corresponds to the C–O stretching vibrations in the phenol group. Finally, the broad peak around 1564 cm$^{-1}$ can be attributed to the C=C stretching vibrations in the aromatic rings.

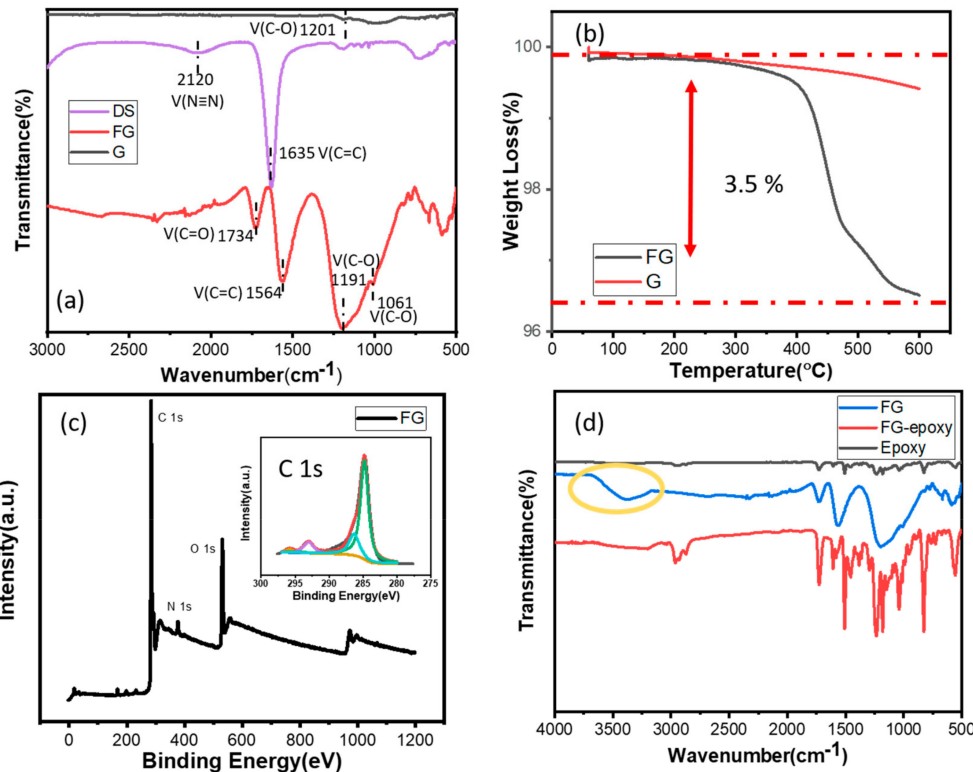

**Figure 3.** (**a**) The infrared spectrum of modified graphene platelets (FG), prepared diazonium salt (DS), and pristine graphene platelet (G); (**b**) TGA data of FG and G; (**c**) XPS spectrum of FG (the inset is the C 1*s* fitting curves); (**d**) infrared spectrum of epoxy, FG, and FG enhanced epoxy coatings.

One possible grafting mechanism is represented in Figure 4 and presumes the presence of oxidation sites/defects on graphene sheets. First, a heteropolar bond between carboxyl and diazonium moieties is formed due to ion exchange, and then the unstable diazonium carboxylate releases nitrogen. This mechanism is similar to the well-known reaction of methylation of carboxylic acids with diazomethane [16].

**Figure 4.** A possible mechanism of graphene functionalization using a diazonium salt.

One may question if the oxidized sites on graphene sheets can be directly reacted with epoxy groups. They cannot since, post-exfoliation, these sites appear to be in a nonreactive ester form, and they only get activated during the Sandmeyer reaction due to saponification with NaOH.

The above findings are further supported by the TGA analysis of G and FG (Figure 3b). TGA curves of unmodified graphene show a weight loss from 50 to 600 °C, with a sharp weight decrease starting

around 200 °C corresponding to the graft elimination from FG. TGA results show 3.5 wt. % of phenolic groups are grafted on the surface of graphene sheets.

XPS measurements give more detail regarding the bonding between graphene and the aminophenol moiety. As shown in Figure 3c, except for carbon, the peaks of N and O can be observed at the same time. Furthermore, the C 1*s* peak can be curve-fitted into three peaks, which are corresponding to C–C (284.8 eV), π–π satellite (293 eV), and C–O (286 eV), respectively, as shown in Figure 3c inset. All this gives solid evidence of successful surface modification of graphene with diazonium salt.

As shown in Figure 3d, the FTIR peak at around 3500 cm$^{-1}$ corresponding to the hydroxyl groups on FG is no longer present in the spectrum of the FG-epoxy coating, indicating the successful combination of the hydroxyl and epoxy groups. This suggests that FG acts as a reactive cross-linker as intended [17].

### 3.2. Tribological Performance of FG Enhanced Epoxy Coating

To investigate the influence of the FG particle size on the rheological properties (sprayability and film levelling) of the coating, three batches of FG material were prepared by mechanical grinding. According to SEM analysis (see Figure 5), the average particle size of FG1, FG2, and FG3 was 3–6 um, 1–3 um, and 0.5–1 um.

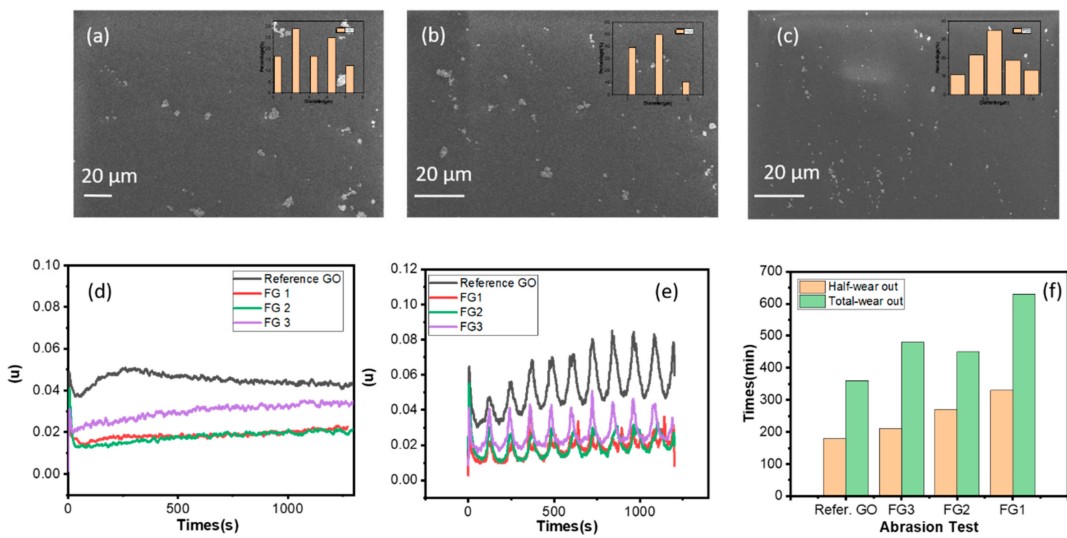

**Figure 5.** (**a**–**c**) SEM images of FG1, FG2, and FG3 (insets are size distributions); (**d**) abrasion tests on stainless-steel plates (4 N, 200 rpm, 30 min, dry contact) of reference GO, FG1, FG2, and FG3; (**e**) friction tests on different coated stainless-steel plates (4 N, 200 rpm, 30 min, lubricated contact) of reference GO, FG1, FG2, and FG3; (**f**) wear data for the reference GO and FG1, FG2, and FG3.

As shown in Figure S1, the structure of reference GO is similar with pristine G. The coefficient of friction (COF) of graphene-free epoxy coating against steel in a dry contact is 0.75, as shown in Figure S2. The same coating using GO filler gives the COF around 0.05, some 15 times less! FG1, FG2, and FG3 show further decrease of COF compared to GO. FG1 and FG2 samples have the lowest COFs, close to 0.02, while FG3 has a slightly higher COF ~0.03.

In a lubricated contact, all the FG samples have lower COF than that of the reference GO sample, see Figure 5e. COF of FG3 is somewhat higher than that of FG1 and FG2.

Relative lifetimes (total wear out) of wear tests are displayed in Figure 5f. FG1 sample has the longest relative lifetime (630 min) which is almost 1.5 times longer than the reference GO coating (360 min).

In summary, both in the friction and wear tests, FG modified coatings demonstrated improved tribological properties. Overall, tribological performance of FG sample is FG1 > FG2 > FG3. This indicates that large size graphene sheets exhibit superior self-lubricating capacity.

### 3.3. Coating Morphology

SEM was used to analyze the coating surface morphology. As shown in Figure 6c,d, the FG flakes are homogeneously dispersed in the epoxy matrix, the average particle size being 1–2 um. Compared with the FG-free epoxy coating, the FG modified coating shows a smoother surface, while the FG-free coating looks more stretched. This indicates the enhanced abrasion resistance of FG modified coatings.

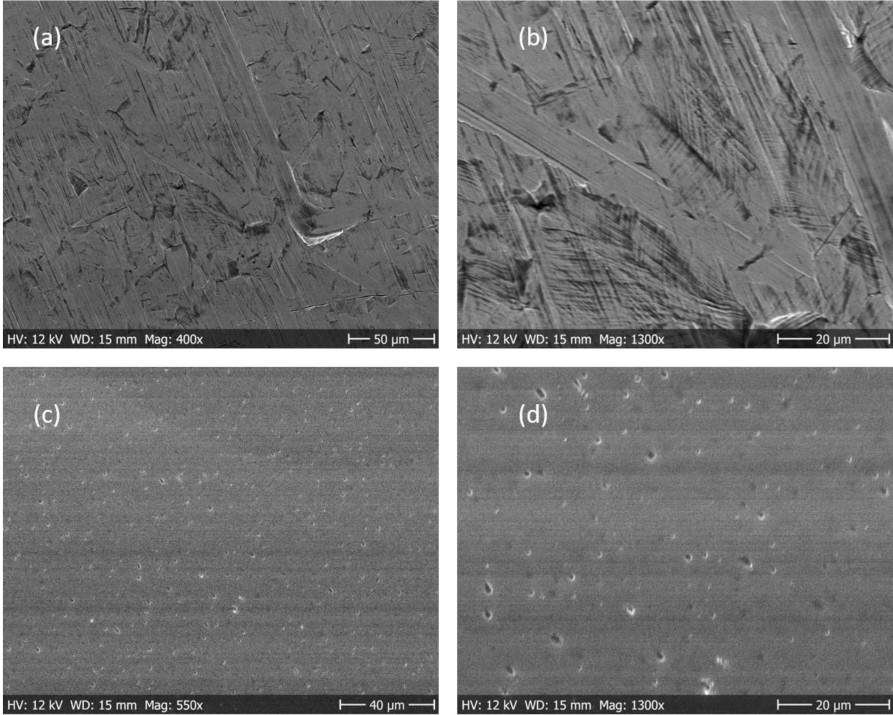

**Figure 6.** (**a**) SEM image of epoxy coating; (**b**) high-resolution SEM image of epoxy coating; (**c**) SEM image of epoxy coating with FG; (**d**) high-resolution SEM image of epoxy coating with FG.

### 4. Conclusions

Mechanically exfoliated graphene was surface modified using an aryl diazonium salt in order to improve its dispersibility in water. Such functionalized graphene was successfully used as reactive filler in a water-borne epoxy coating formulation. Coatings modified by functionalized graphene demonstrated improved tribological properties in terms of friction and wear. This work paves the way for future applications of functionalized graphene for mechanical enhancement of polymer-bonded coatings.

**Supplementary Materials:** The following are available online at http://www.mdpi.com/2079-6412/9/11/754/s1: Figure S1, (a) SEM images of the pristine graphene (the inset shows its size distribution), (b) SEM image of the reference GO (the inset shows its size distribution), (c) the infrared spectrum of GO, (d) raman spectrum of GO; Figure S2, Abrasion test on stainless-steel plates of pristine epoxy binder (4N, 200rpm, 30min, dry contact).

**Author Contributions:** Conceptualization, Y.L., L.Y., J.L., N.W., C.X. and B.Z.; Methodology, Y.L., C.X. and B.Z.; Software, Y.L.; Validation, Y.L., C.X. and L.Y.; Formal analysis, Y.L. and C.X.; Investigation, Y.L., L.Y., J.L., C.X. and B.Z.; Resources, J.L., L.Y., B.Z. and H.L.; Data curation, Y.L.; Writing—original draft preparation, Y.L., C.X. and A.Z.; Writing—review and editing, J.L., B.Z. and H.L.; Visualization, J.L. and B.Z.; Supervision, J.L., L.Y., B.Z. and H.L.; Project administration, L.Y.; Funding acquisition, J.L., L.Y., B.Z. and H.L.

**Funding:** This research was funded by the Swedish Board for Innovation (Vinnova) under the Sioagrafen program, by the Swedish Board for Strategic Research (SSF) with the contract No: SE13-0061 and GMT14-0045, by Formas with the contract No: FR-2017/0009 and by STINT for the double degree PhD collaboration program with the contract No: DD2016-6502, by the Swedish National Science Foundation with the contract No: 621-2007-4660 as well as by the Production Area of Advance at Chalmers University of Technology, Sweden. J.L. also acknowledges the financial support by the Key R&D Development Program from the Ministry of Science and Technology of China with the contract No: 2017YFB0406000 as well as from the National Natural Science Foundation of China (No: 51872182). H.L. thanks financial support from Shanghai International Collaboration research project (19520713900) and State Key Laboratory of Molecular Engineering of Polymers in Fudan University.

**Conflicts of Interest:** The authors declare no conflict of interest.

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
