# Peer review of "Surface Modification of Graphene for Use as a Structural Fortifier in Water-Borne Epoxy Coatings"

_coatings, doi:10.3390/coatings9110754_

Round 1
Reviewer 1 Report
1) The section "Materials and Methods" is designed carelessly. The characteristics of all used and reagent materials, including purity and cleaning methods, if applicable, should be specified in detail. I could not find any characteristics of unmodified graphene, as well as epoxy.
2) What is the yield of modified graphene?
3) The arguments about the mechanism of functionalization of graphene should be transferred to the section “Results and discussions”. At the same time, this mechanism makes us think about the feasibility of carrying out the modification proposed by the authors. I agree that phenolic groups at high temperature (for example, 250 C, as in this article) will react with epoxy. But I do not understand what is the point of their introduction, because carboxylic groups, the presence of which the authors suggest, also react well with epoxy and are more hydrophilic than phenolic. This means that the graphene surface should already be hydrophilic enough to form stable dispersions and be wetted by an epoxy binder. Then what is the point of such a modification? This point (and, probably, also the mechanism of functionalization of graphene) must be explained in details.
4) Figure 3b is visible in graphene layers prior to functionalization. Why do not the authors cite a similar micrograph after modification?
5) The authors showed that functionalized graphene loses weight starting at 200 degrees C, then why cure the coating at 250 degrees C?
6) On the IR spectra, the absorption bands discussed in the text should be indicated and designated.
7) In fig. 5 no image of the coating with unmodified graphene. What is its particle size?
8) The article can be published in the Coatings journal after the correction of the listed defects.
Reviewer 2 Report
English needs to be improved through out the document
Improper controls or no control data shown in spectroscopic and thermal analysis.
Confusing and improperly labeled figure captions
Reviewer 3 Report
In my opinion this paper should be comprehensively reviewed by authors.
My main comments:
introduction: Based on references or own experience please explain why diazonium salt was used as modifier of graphene surface. -> highlight novelty of work characterization part: Please provide more details about measurements conditions for each method. Figure 3d - I did not observed layers in case of FG, please mark them in figure. Figure 4 - please explain and mention in text why the FTIR spectra of pure graphene is flat. Please present all FTIR (Figure 4a) in full range: 4000-500 cm-1. TGA results - please compare this results with theoretical value based on the applied content of modifier and comment on observed differences. Figure 5 - provide SEM images for reference filler and information about its avarage particle size. Provide standard deviation for presented resultsAuthor Response
Please see the attachment.

Round 2
Reviewer 1 Report
Now I can recommend this paper to be published in Coatings.
However, there are still some minor corrections to be done.
Line 89 – please change “The yield of p-hydroxybenzene diazonium chloride was around 80%” by “The yield of p-hydroxybenzene diazonium chloride was 80%” Line 97 “is” should be changed to “was” The English language should be checked carefully in the entire manuscript, including lines 45, 54, 133-134, 65-68, these sentences may be improved. In 2.1 it should be noted whether materials were used with or without additional purification. Please also change “Aminophenol” by “p-aminophenol”. Please change Sandmayer by Sandmeyer.
Author Response
Thanks for the comments on the manuscript. We corrected errors pointed out in your suggestion as below.
The yield of p-hydroxybenzene diazonium chloride was 80%.
By analogy with the Sandmeyer reaction, 2 g p-aminophenol was added to 100 mL 5 % sodium hydroxide solution at 50 °C and stirred for 30 min.
According to your suggestion, we have polished and rewrote the whole manuscript. Please check more language modification in attached manuscript.

Reviewer 2 Report
Extensive English revision is still required throughout the manuscript. Some of the newly added content also has poor English word choice and sentence structure. I highly recommend the use of an editing service to work through the many difficult sentences. I feel I must make this stand as too many manuscripts have been allowed to be published in this state greatly diminishing the body of literature.
The FTIR spectra shown are not normalized so all peaks can be compared between samples in the absence of specimen artifacts. A good example of this is the epoxy spectra in Fig. 3d. Its suppressed state makes comparisons impossible. This fact weakens your supposition that the absence of the hydroxyl peaks proves reaction. Changes in Tg or yield could also be used as evidence.
Reviewer 3 Report
All my comments were included in revised paper.
I found some typos and minor errors, for example:
page 9/262 (space between 15. and cited paper) page 4/147 (different font format)Please check this during proofs-reading. I recommend publication of this paper.
Author Response
Thanks for the kindly suggestion and approve for publication. We have corrected the stylish error as suggested. Please check it as below.
Kuila, T.; Bose, S.; Mishra, A. K.; Khanra, P.; Kim, N. H.; Lee, J. H., Chemical functionalization of graphene and its applications. Mater. Sci. 2012, 57 (7), 1061-1105. The above findings are further supported by the TGA analysis of G and FG (Fig. 3(b)). TGA curves of unmodified graphene shows a slow weight loss from 50 to 600 °C.
